# Chemical Composition and Antimicrobial Potential of a Plant-Based Substance for the Treatment of Seborrheic Dermatitis

**DOI:** 10.3390/ph16030328

**Published:** 2023-02-21

**Authors:** Viktor A. Filatov, Olesya Yu. Kulyak, Elena I. Kalenikova

**Affiliations:** 1Department of Pharmaceutical Chemistry, Pharmacognosy and Organization of Pharmaceutical Business, Faculty of Basic Medicine, M.V. Lomonosov Moscow State University, 27/1 Lomonosovsky Avenue, 119991 Moscow, Russia; 2SkyLab AG, 1066 Lausanne, Switzerland; 3All-Russian Scientific Research Institute of Medicinal and Aromatic Plants, 117216 Moscow, Russia

**Keywords:** antimicrobial, antifungal, essential oils, terpenes, GC/MS, seborrheic dermatitis

## Abstract

Seborrheic dermatitis (SD) is the most prevalent dermatological disease, occurring in up to 50% of newborns, children, and adults around the world. The antibacterial and antifungal resistance contributed to the search for new natural substances and the development of a novel substance based on *Melaleuca alternifolia* (*M. alternifolia*) leaf oil (TTO), 1,8-cineole (eucalyptol), and α-(-)-bisabolol. Thus, this work aimed to determine the chemical composition of the novel plant-based substance and to evaluate its antimicrobial activity against standard microorganisms involved in the pathogenesis of SD. Moreover, the chemical composition of the substance was analyzed by gas chromatography coupled with mass spectrometry (GC/MS). *Staphylococcus epidermidis* (*S. epidermidis*), *Staphylococcus aureus* (*S. aureus*), *Micrococcus luteus* (*M. luteus*), and *Candida albicans* (*C. albicans*) were used for antimicrobial and antifungal assays by means of the broth microdilution method to determine the minimal inhibitory concentration (MIC). Finally, the substance’s ability to inhibit *Malassezia furfur* (*M. furfur*) was evaluated. Eighteen compounds from different chemical groups were identified by GC/MS. The major biologically active compounds of the substance were terpinen-4-ol (20.88%), 1,8-cineole (22.28%), (-)-α-bisabolol (25.73%), and o-cymene (8.16%). The results showed that the substance has a synergistic antimicrobial and antifungal activity, while *S. epidermidis* and *C. albicans* strains were the most susceptible. Furthermore, the substance inhibited *M. furfur*, which is a main pathogen involved in the pathogenesis of SD and clinical manifestations. It can be concluded that the novel plant-based substance has a promising potential against *M. furfur* and scalp commensal bacteria and may be helpful for the development of new drugs for treatment of dandruff and SD.

## 1. Introduction

Dermatological diseases of the scalp have a high prevalence worldwide. It is established that seborrheic dermatitis (SD) is the most common dermatological disease of the scalp, occurring in up to 50% of adult population around the world and 42% of newborns [1]. The detailed etiopathogenesis of SD is unknown and required additional research and clinical studies [2]. Various intrinsic and environmental factors, such as sebaceous gland secretion, changes in microflora diversity, fungal colonization, individual susceptibility, sexual hormones, poor hygiene of the scalp, stress, improper use of shampoo, and other cosmetics can affect SD [2]. Besides the physical discomfort such as itching, excessive sebum production, and a dry scalp, seborrheic dermatitis is socially embarrassing and negatively impacts patients’ well-being [3].

SD can be present with superficial perivascular and perifollicular inflammatory infiltrates, parakeratosis, inflammation surrounding hair follicles, and itchy and excessive quantities of sebum. The treatment of SD and dandruff focuses on ameliorating symptoms and maintaining remission with long-term therapy but only targets fungal colonization and inflammation. The most common treatments are based on synthetic topical antifungal and anti-inflammatory agents in accordance with clinical recommendations of different countries. The application of these synthetic agents is useful for the effective treatment of SD, but regular use of products with these substances can change the scalp microflora biodiversity [4] and is determined to be high risk of further antimicrobial resistance [5]. The long-term application of antifungal and anti-inflammatory substances causes immunodeficiency, skin atrophy, telangiectasia, skin infections, skin discoloration, and hypopigmentation of the scalp [6]. Contact dermatitis, itching, a burning sensation, and a dry scalp after drug treatment occurs in 3% of patients with SD [7,8].

Antifungal resistance is a common serious public health concern around the world, being considered as a global emerging challenge in the treatment of dermatological diseases associated with resistant fungi and bacteria. This issue has been intensified due to the prolonged and uncontrolled use of antimicrobials and antifungals, which has caused the adaptation of microorganisms [9] such as *Malassezia species*. *Malassezia* strains involved in the pathogenesis of SD have become resistant to a topical antifungal treatment, resulting in the treatment not being clinically efficacious in up to one-third of patients with diagnosed SD [10]. Antifungals based on azoles are recommended as the first-choice antifungal treatment for *Malassezia*-associated diseases such as dandruff and SD, but these drugs are less effective due to the decreased sensitivity of *Malassezia species*. These mechanisms are related to genetic mutations in the cytochrome P450 lanosterol 14α-demethylase (ERG11/CYP51) gene, the increased expression of the transporters of the ATP-binding cassette (ABC) as *Candida* drug resistance 1 protein (CDR1p) or *Malassezia* drug resistance protein (MDR), and drug efflux [11]. The disease-derived isolates of *Malassezia furfur* (*M. furfur*) have become less sensitive to terbinafine, clotrimazole, ketoconazole, amphotericin B, and miconazole treatment, which has led to an increase in the minimum inhibitory concentrations (MICs) of these drugs [11]. Genes involved in metabolism and secondary metabolite production showed increased expression and upregulation in the disease strains of *M. furfur* after long-term exposure to azole antifungals [11]. The spread of resistant microorganisms has been a motivational factor into the search for new effective alternatives from plants [12]. Natural compounds such as different plant extracts and essential oils (EOs) have been viewed as a priority for the investigation of their potential roles as effective antimicrobials and antifungals, due to their composition of biologically active compounds, multiple properties, and different activities [13] that can be helpful for the treatment of SD.

An epidemiological situation can result in a new surge in the interest in prospective plant-derived pharmaceutical substances for proper scalp treatment. The development of new plant-derived pharmaceutical substances is necessary to eliminate the causes and consequences of dandruff and SD. Research into the pathogenesis of dandruff and seborrheic dermatitis has led to new prophylaxis and treatment approaches [2]. New clinical studies have shown that microflora diversity determines a healthy scalp and plays a key role in the pathogenesis of SD [14]. Therefore, the chosen therapy should not affect bacteria such as *Staphylococcus epidermidis* (*S. epidermidis*), *Staphylococcus aureus* (*S. aureus*), and *Cutibacterium acnes* (*C. acnes*). The commensal bacteria *C. acnes* help to maintain an acidic environment to decrease trans epidermal water loss (TEWL) and decrease the prevalence of *Malassezia species* [15]. An increase in the diversity of *S. epidermidis* leads to the secretion of cytokines and the involvement of immune cells in the pathogenesis that complicates SD [16]. In this regard, patients with SD require treatment based on new active substances that are safe, effective, and plant-based, and that exhibit low irritancy potential, are skin-friendly (do not disrupt the skin microflora of the scalp), and do not cause rapid antimicrobial resistance and/or balance its diversity. One such group of substances includes biologically active substances of natural origin from EOs or plant sources.

EOs of plant origin have gained global attention in dermatology. EOs are defined as plant mixtures of volatile aromatic compounds from different chemical classes that are poorly soluble in water and extracted from different plant parts through steam distillation or a cold-pressing process [17]. EOs contain high levels of aromatic secondary metabolites that belong to terpenes and phenolic components, provide a pleasant smell, and have useful antimicrobial and antifungal properties [18], in addition to their anti-inflammatory [19] and antioxidant [20] properties, among others [21].

It was discovered by the authors that the most promising substances for the treatment of seborrheic dermatitis are *Melaleuca alternifolia* (*M. alternifolia*) leaf oil (TTO), 1,8-cineole (eucalyptol), and (-)-α-bisabolol. The TTO has a comprehensive chemical composition and a number of antimicrobial and antifungal activities [22]. The main biologically active compounds of TTO are terpenes and terpenoids such as terpinene-4-ol, terpinolene, α-terpinene, and 1,8-cineole. It was discovered that TTO has antibacterial, antifungal, antiviral, and antiprotozoal activity [22,23,24,25,26,27]. Numerous studies have shown anti-inflammatory activity of TTO in vitro and in vivo [19,28,29,30] that can help reduce inflammation of the scalp in the case of SD. Eucalyptol—1,8-cineole (1,3,3-trimethyl-2-oxabicyclo [2.2.2]acetate) is a main monoterpenoid of essential oils obtained from various *Eucalyptus* species [31] and has a mint-like smell. The compound has antibacterial, antifungal, anti-inflammatory, and cough-suppressant properties [31,32,33]. Previous studies have shown that 1,8-cineole acts as a permeability enhancer and can improve the antimicrobial activity of drugs against different bacterial strains [34,35]. (-)-α-Bisabolol belongs to the group of monocyclic sesquiterpenes derivatives and can be obtained from plant resources or by chemical synthesis that is a critical point for molecule’s optical rotation and biological activity. This active compound is widely used in cosmetics and dermatological drugs due to its antibacterial, skin repairing, antiallergic, wound healing, and anti-inflammatory properties, in combination with its generally recognized as safe (GRAS) status [36]. Its biological activity is determined by a high lipophilicity that changes the structure of the bacterial and fungal cell wall with subsequent lysis, and helps to enhance skin absorption of some molecules for deeper action in the scalp [22,37].

The plant-based substance TTO, 1,8-cineole, and (-)-α-bisabolol could have antibacterial activity on skin residential microorganisms such as *S. epidermidis* and *S. aureus* [22,34,38], which are involved in the pathogenesis of SD. Due to the heterogenous composition [39] of the substance based on TTO, 1,8-cineole and (-)-α-bisabolol, in addition to unanswered questions concerning its stability and possible reactions when the three components are combined [40], there is the need to determine the content of phytochemicals in the mixture and characterize the plant-based pharmaceutical substance specification. To the best of our knowledge, gas chromatography/mass spectrometry (GC/MS analysis) is the most suitable method for the investigation of the chemical composition of volatile compounds from EOs, offering information regarding identity, possible impurities, and new phytoconstituents [40,41].

In this context, it was hypothesized that the novel plant-based substance based on biologically active ingredients could exhibit antimicrobial and antifungal activity against several standardized strains of the pathogenic fungi and bacteria involved in the pathogenesis of SD. In order to test the hypothesis, this work aimed to determine the chemical composition of the novel plant-based substance based on TTO, 1,8-cineole, and (-)-α-bisabolol, as well as to verify its antimicrobial and antifungal potential against standard microorganisms involved in the pathogenesis of SD.

## 2. Results and Discussion

### 2.1. GC/MS Analysis of TTO and the Plant-Based Substance

The chemical composition of TTO as the abundant phytoconstituent of the plant-based substance comprised fifteen bioactive compounds (Figure 1). These compounds have characteristic peaks and mass spectra that were compared along with their analogs reported in the National Institute of Standards and Technology-2017 (NIST-2017) and Willey-08 libraries.

TTO contained terpenes with characteristic chemical structures; therefore, a full qualitive composition was needed for standardization. The major components of TTO such as α-terpinene, γ-terpinene, α-terpineol, and terpinen-4-ol (Figure 2) are specific to TTO and have antibacterial activity [37]. In accordance with the international standard ISO 4730, standardization of TTO was carried out according to the content of terpinen-4-ol, the content of which should be at least 30% [43].

The mass spectrum of terpinen-4-ol is shown in comparison with a substance from the NIST-2017 library (Figure 3). TTO is a promising essential oil in the prevention and treatment of dandruff and SD pathogenesis [44]. This oil has a plenty of volatile aromatic secondary metabolites belonging to terpenes and phenolic chemical classes with established antimicrobial and antibiofilm properties [18]. Alternative therapies, including TTO, have been reported to demonstrate moderate antimicrobial activity by disrupting the microbial growth associated with dandruff and SD pathogenesis [44]. It was found that the TTO minimum inhibitory concentrations (MICs) of fluconazole-resistant fluconazole-susceptible isolates were comparable [45] due to the high content of terpinen-4-ol [46] as a main phytoconstituent of TTO.

In addition, TTO contains 1,8-cineole in an amount of 5 to 10% as a secondary metabolite to potentiate the action of other components [47]. The mass spectrum of 1,8-cineole is shown in comparison with a substance from the NIST-2017 library (Figure 4). The 1,8-cineole compound can be used for the treatment of SD and dandruff in cases of azole-resistant human pathogenic *Malassezia* and other fungi species [46]. It acts as an enhancer for the high permeability of the phytoconstituents throughout the bacterial and fungal cell wall. The lipophilic properties of 1,8-cineole determine its activity, such as the increase in membrane permeability and fluidity, impaired enzyme function in the cell wall, and ion transfer that lead to cell death [48]. The combination of terpinene-4-ol and 1,8-cineole has an additive antibacterial effect against Gram-positive and Gram-negative microorganisms [49]. In addition, 1,8-cineole exhibits anti-quorum-sensing activity because of the decrease in the secretion of quorum-sensing molecules and the prevention of biofilm formation [50].

According to the phytochemical analysis of the substance based on TTO, 1,8-cineole and (-)-α-bisabolol (ratio 1:1:1), eighteen chemical phytoconstituents were identified, representing 98.64% of the total composition. We found three nonidentified compounds (Figure 5) that could be related to the main compounds of the plant-based substance.

Most of identified phytoconstituents were monoterpenes (67.09%) and sesquiterpenes (31.55%). Moreover, the most abundant constituents of the analyzed the plant-based substance were (-)-α-bisabolol (25.73%), 1,8-cineole (22.28%), and terpinene-4-ol (20.89%). These compounds (Figure 6) should be added to the technical specification for the description and quality assessment of the plant-based substance.

The mass spectrum of (-)-α-bisabolol is shown in comparison with a substance from the NIST-2017 library (Figure 7). The (-)-α-bisabolol compound is a sesquiterpene that exhibits anti-inflammatory and antibacterial activities [51,52]. This compound decreases the secretion of arachidonic acid metabolites and thereby reduces high sensitivity [52], which can be helpful for the treatment of SD and dandruff. Furthermore, (-)-α-bisabolol acts as an enhancer for the transdermal delivery of phytoconstituents, resulting in a deeper action in the skin [53]. The high antifungal activity of (-)-α-bisabolol is compatible with azelaic acid and is useful in the treatment of dermatomycosis [54].

The phytoconstituents o-cymene (8.16%), γ-terpinene (6.09%), α-terpineol (2.96%), and farnesol (2.16%) also showed reasonable concentrations. The α-pinene, β-pinene, α-terpinene, o-cymene, 1,8-cineole, γ-terpinene, α-terpinolene, terpinen-4-ol, α-terpineol, and δ-cadinene compounds were identified in TTO, as well as in different parts of *M. alternifolia* [55]. Regarding the phytochemicals with the lowest concentration traces (<1%), 1-methyl-4-(1-methylethenyl)-2-cyclohexene-1-ol (0.36%), α-phellandrene (0.32%), β-pinene (0.26%), and sabinene (0.20%) were identified (Table 2).

Mass spectrometry (MS) provides information on all components and trace amounts of substances in complex multicomponent mixtures such as EOs. This method is highly sensitive for the identification of substances by comparison with mass spectra libraries. The main characteristics of the mass spectra for constituents in the plant-based substance evaluated is shown in comparison with the NIST-2017 library (Table 3). The experimental *m/z* peaks were identical to the comparison peaks in the NIST-2017 library, providing evidence for the accuracy of the method and chosen conditions for GS/MS.

Furthermore, the qualitative analysis confirmed the presence of ten new phytoconstituents: 1-methyl-4-(1-methylethenyl)-2-cyclohexene-1-ol, trans-ascaridole glycol, sabinene, α-phellandrene, trans-caryophyllene, farnesol, alloaromadendrene, and three nonidentified compounds with a terpene structure. The monoterpenes sabinene and α-phellandrene were not initially found in the TTO but were detected in the novel plant-based substance in a minor percentage not exceeding 0.20% and 0.32%, respectively. GC/MS revealed the trans-caryophyllene and farnesol that could have originated from the (-)-α-bisabolol sample. It is estimated that the farnesol concentration was present up to 2.16%, and that of trans-caryophyllene was up to 1.08%. Identified compounds such as 1-methyl-4-(1-methylethenyl)-2-cyclohexene-1-ol and trans-ascaridole glycol were structurally related to 1,8-cineole and were presumably generated in the plant-based substance due to the presence of (-)-α-bisabolol, its related phytoconstituents, and some biologically active compounds of TTO.

Moreover, 1-methyl-4-(1-methylethenyl)-2-cyclohexene-1-ol, as a new derivative in the substance, was previously detected in the chemical composition of *Achillea santolina* L. essential oil and demonstrated effectiveness against microorganisms and leishmania [56]. The compound trans-ascaridol, which was also identified amongst the main constituents of the Indian *Chenopodium ambrosioides* and *Chenopodium botrys*, has been found to have antifungal activity [57]. Interestingly, alloaromadendrene has been reported to have a higher antimicrobial activity than 1,8-cineole and to increase its antimicrobial potential [58]. Additionally, this compound causes the disruption of microbial membranes due to its high lipophilicity [59,60].

The GC/MS analysis of the novel plant-based substance evaluated in this study detected the highest concentrations of (-)-α-bisabolol (25.73%), 1,8-cineole (22.28%), terpinen-4-ol (20.89%), and other phytochemicals. Overall, the full chemical composition and characteristics of the plant-based substance are presented in Table 2. Even minor or trace constituents could play a role as enhancers of antibacterial activity and possibly produce synergistic effects with major components [61]. The TTO with a rich composition [62], 1,8-cineole, and (-)-α-bisabolol are responsible for the antimicrobial and antifungal activities that can be useful for the treatment of mild and moderate SD. Each compound has an additional role in the plant-based substance. The TTO-enriched terpinen-4-ol, 1,8-cineole and (-)-α-bisabolol can have a hypothetical synergistic effect [49,50] and good membrane permeability into bacteria and fungi [63] related to SD. Additionally, the plant-based substance can have activity against the biofilms and the quorum sensing of microorganisms that are needed for scalp microflora regulation [50].

### 2.2. Antimicrobial Activity

The influence of each bioactive compound and the plant-based substance was estimated using a broth microdilution method against the standardized reference strains *S. aureus* ATCC 29213, *S. epidermidis* ATCC 14990, *C. albicans* ATCC 10231, and *M. luteus* ATCC 10240a. According to the results described in Table 4, the compounds showed MIC values from 1.25 to 40.00 mg/mL for different strains, while the MIC values of benzalkonium chloride used as the active control were 5 to 10 mg/mL.

TTO showed MIC values of 40 mg/mL against *S. aureus* and *M. luteus* but had a higher activity with MIC values of 1.25 and 5.00 mg/mL for the *S. epidermidis* and *C. albicans* strains, respectively. The high antibacterial activity of 1,8-cineole was observed only for *S. epidermidis*. Other microbial strains showed a low sensitivity to 1,8-cineole. The (-)-α-bisabolol component exhibited antibacterial activity against *S. aureus* and *S. epidermidis*, while *M. luteus* and *C. albicans* were not very sensitive in presence of (-)-α-bisabolol. The active control benzalkonium chloride showed moderate activity against all microbial strains because of its broad-spectrum antimicrobial activity

The plant-based substance based on TTO, 1,8-cineole, and (-)-α-bisabolol showed an MIC value of 1.25 mg/mL against all tested microorganisms except *M. luteus*. A synergistic antimicrobial activity of combining three compounds was observed, showing low activity against scalp- and skin-beneficial *M. luteus*. It was verified that the plant-based pharmaceutical substance based on TTO, 1,8-cineole, and (-)-α-bisabolol in a ratio of 1:1:1 had an improved antimicrobial activity against *S. aureus*, *S. epidermidis*, and *C. albicans* involved in the pathogenesis of SD that could be useful for the prophylaxis and treatment of SD and the targeted regulation of scalp microflora for the maintenance of a healthy scalp appearance in cases of SD.

### 2.3. Antifungal Activity against M. furfur

An in vitro comparative evaluation of the antifungal activity of substance vs. control in terms of the MIC determination (Table 5) was performed via the colony counting method. The reference microbial strains, namely, *M. furfur* ATCC 14251 and *C. albicans* ATCC 10231, were selected for the experiment. It was found that the plant-based substance showed high antifungal activity against *C. albicans* (MIC = 0.25%) and *M. furfur* (MIC = 0.50%) at concentrations lower than the positive control, climbazole (MIC = 1.00%). Ketoconazole, a standard targeted drug widely prescribed and used in the treatment of SD, showed the highest antifungal activity (MIC = 0.25%) against *M. furfur* and *C. albicans*. Nevertheless, resistance of fungi to azole drugs such as ketoconazole is expected to increase in the nearest future [11,64] due to mutations, gene upregulation, and fungi adaptations. Therefore the plant-based substance based on TTO, 1,8-cineole, and (-)-α-bisabolol in a ratio of 1:1:1 presented an MIC of 0.50% for *M. furfur*, had a high antifungal potential against fungi involved in the pathogenesis of SD, and could be used for the development of new therapeutic agents for the treatment of SD and other fungal skin infections after full research of its toxicity, dermatological tolerance, and extended clinical studies.

According to the obtained results, the plant-based substance inhibited the growth of the standard strains such as *C. albicans* and *M. furfur* obtained from ATCC. Although the antifungal activity and reduction in colony-forming units (CFU) seemed to be determined by the final concentration of the substance and the tested fungal strain, the plant-based substance demonstrated a decrease in more than *2*log10CFU of M. furfur (Table 6), corresponding to >99% antifungal activity against the chosen *Malassezia* strain.

The plant-based substance based on TTO, 1,8-cineole, and (-)-α-bisabolol, exhibited a pronounced antifungal effect against *M. furfur*, resulting in a decrease in fungi after 1 h of contact. It was shown the significant inhibition in *M. furfur* growth in the presence of the plant-based substance and with an average efficiency of 99.75% (equivalent to a 2.6 log10CFU reduction) at a concentration of 0.50%. The plant-based substance at the maximum tested concentration of 1.00% showed a reduction of 2.81 log10CFU, that implied a reduction of 99.85% against *M. furfur* and could be sufficient for the treatment of SD.

Ketoconazole was the active control in the experiments using the *M. furfur* reference strain. No fungal growth was found after addition and cultivation with ketoconazole at a concentration above 0.25% (Table 6). Ketoconazole, as the gold standard in the treatment of SD, is used in medical shampoos; however, its activity has decreased due to the subsequent resistance of *Malassezia* fungi [11]. Climbazole demonstrated dose-dependent inhibition in *M. furfur* growth; however, a log10CFU more than 2.0 was not achieved, even at a high concentration of 1.00% (Table 6). Consequently, it cannot be considered a first-choice drug for the treatment of SD. Moreover, ketoconazole and climbazole had only specific antifungal activity, while antimicrobial activity against *S. aureus* and *S. epidermidis* involved in the pathogenesis of SD was insufficient. New data show that new drugs for the treatment of SD should be effective against both fungi and bacteria [14,16]; this is the main disadvantage of recommended therapies for SD.

Thus, the heterogenous chemical composition of the plant-based substance based on TTO, 1,8-cineole, and (-)-α-bisabolol was evaluated via GC/MS analysis. Based on the results, the composition was rich in monoterpenes and sesquiterpenes. These biologically active phytoconstituents have a range of activities such as antimicrobial [65], anti-inflammatory [19,28,29,30,31,32,33], and antibiofilm [34,35,50] properties, skin southing [36], and membrane permeability enhancement [22,66,67]. The relative richness of the composition of the plant-based substance demonstrated complex antibacterial and antifungal activity against ATCC strains chosen in accordance with their involvement in the pathogenesis of SD. The experiment showed the synergistic antimicrobial activity of the plant-based substance compared to single compounds and the active control such as benzalkonium chloride. A bacterial strain such as *M. luteus* had a low sensitivity to the plant-based substance that could be beneficial strategy for a scalp-microbiome-friendly approach and checked as a part of expanded clinical trials. Previous studies [22,65,66,67,68,69] and current research corroborated that the addition of 1,8-cineole and (-)-α-bisabolol can induce the antimicrobial activity of TTO and act as penetration enhancers with other beneficial properties for the treatment of SD. The combined plant-based substance at a concentration of 0.5–1.0% caused a marked inhibition in the fungal growth of *M. furfur*, showing a significant log10CFU reduction equivalent to an average efficiency of 99.75–99.85%. This effect is higher than the climbazole-mediated log10CFU reduction and compatible with ketoconazole efficiency. The recommended antifungal drugs based on the azole structure revealed decreased activity due to subsequent resistance of *Malassezia* fungi [11] that may limit their use in the nearest future. There are no data about the resistance of *Malassezia* species to the main compounds of the plant-based substance—TTO, 1,8-cineole, and (-)-α-bisabolol —that could be used in the prospective treatment of dandruff and SD. Research on the toxicological profile, dermatological tolerance, and clinical efficacy of developed dermatological drugs with this plant-based substance is needed for a further evaluation of possible and unknown properties.

## 3. Materials and Methods

### 3.1. Chemicals and Plant Materials

*M. alternifolia* leaf oil (CAS 68647-73-4), 1,8-cineole (CAS 470-82-6), and α-(-)-bisabolol (CAS 23089-26-1) were purchased from Sigma-Aldrich (Sigma Chemical Co. Ltd., St. Louis, MO, USA) and were prepared using standard methods before use. The substances had a purity of at least 99.0%. The plant compounds selected for analysis are presented in Table 7. Technical-grade chloroform was used for GS/MS analysis as a solvent.

### 3.2. Analysis of the Chemical Composition of the Substance

The chemical analysis of TTO and the plant-based substance based on TTO, 1,8-cineole, and (-)-α-bisabolol at a ratio of 1:1:1 was performed via GC/MS. The three phytoconstituents were taken in an equal proportion of 1:1:1 by weight for the determination of the type of antimicrobial activity, to be compared with the GC/MS of pure substances [69].

The GC/MS analysis was conducted using an Agilent Technologies 6890N gas chromatograph coupled with a 7000 mass spectrometer (Agilent, Santa Clara, CA, USA) and an HP-5 MS low-bleed capillary column (20 m × 0.18 mm i.d., 0.18 μm film thickness (Agilent Technologies, Santa Clara, CA, USA)). The sample for analysis was prepared by the addition of 50 μL of the TTO to 1 mL chloroform. An injection volume of 1 μL was subjected to a splitless mode (1:30) during the GC/MS analysis. Helium was used as the carrier medium at a constant flow rate of 1.25 mL/min. The oven temperature was programmed as follows: maintained at 80 °C for 1 min, then programmed to increase to 140 °C at 10 °C/min, raised to 280 °C at 20 °C/min, and finally kept at 280 °C for 20 min. The MS was programmed as follows: ion source temperature at 230 °C, interface temperature at 280 °C, ionization voltage at 70 eV, and a scan range of *m/z* 29 to 500 amu. The chemical identification was performed by comparing the obtained phytoconstituent profiles with the corresponding reference retention indices (RIs) and the mass spectra in the NIST-2017 (National Institute of Standards and Technology, Gaithersburg, MD, USA) and Wiley-08 (Wiley, New York, NY, USA) databases, and the Adams libraries [42,70,71], as well as by comparing their GC/MS characteristics with the available published data [72,73,74]. These analyses were carried out in triplicate.

The analysis of the plant-based substance based on TTO, 1,8-cineole, and (-)-α-bisabolol in a ratio of 1:1:1 was performed using the same procedure, with some modifications, for the appropriate separation and determination of the phytoconstituents [41]. The used gas chromatograph was equipped with an HP-5 MS low-bleed capillary column (30 m × 0.25 mm i.d., 0.25 μm film thickness (Agilent Technologies, Santa Clara, CA, USA)). The sample for analysis was prepared by addition of 50 μL of the plant-based substance to 1 mL chloroform. An injection volume of 1 μL was subjected to a splitless mode (1:30) during the analysis. Helium was used as the carrier gas at a constant flow rate of 1.25 mL/min. The oven temperature regimen was the same: maintained at 70 °C for 1 min, then programmed to increase to 290 °C at 10 °C/min, and finally kept at this temperature for 20 min. The MS was programmed with the ion source temperature at 230 °C, the interface temperature at 280 °C, the ionization voltage at 70 eV, and a scan range of *m*/*z* 29 to 500 amu. The relative abundance (%) of the chemical phytoconstituents was determined using the peak area. The chemical detection and determination of found compounds was performed by comparing the respective reference retention indices (RIs) and the mass spectra in the databases, and the spectral literature according to Adams [42]. The available NIST 2017 and Wiley-08 databases were used for this chemical analysis. These analyses were carried out in triplicate.

### 3.3. Antimicrobial Activity

#### 3.3.1. Test Microorganisms and Growth Conditions

The TTO, the plant-based substance based on TTO, 1,8-cineole, and α-(-)-bisabolol in an equal ratio of 1:1:1, and other substances were tested against several bacterial and fungal strains from ATCC that were chosen in accordance with their potential role in the pathogenesis of SD. The tested microorganisms were obtained from the ATCC: *S. epidermidis* ATCC 14990, *S. aureus* ATCC 29213, *M. luteus* 10240a, and *C. albicans* ATCC 10231. The selected microorganism species were cultivated at 37 °C for 24 h under aerobic conditions in Mueller Hinton broth and tryptone soy agar (TSA; Graso Biotech, Starogard Gdański, Poland) [75,76].

#### 3.3.2. Antimicrobial Screening by Determining the MICs for the Substances

The broth microdilution method was used for the determination of MICs for the tested substances. Experiments were performed according to the European Committee on Antimicrobial Susceptibility Testing (EUCAST) rules for antimicrobial susceptibility testing [77]. The 96-well microtiter plates (96 Well EDGE Cell Culture Plates, Nest Scientific Biotechnology, Wuxi, China) were used for consecutive dilutions. The water solubility of the biologically active compounds was improved by adding of 0.3% Polysorbate 20 to initial solutions. Microbial growth controls and sterility controls of the culture medium were performed. The serial dilutions of the compounds were received ranging from 20 to 1.25 mg/mL. Benzalkonium chloride, a well-known antimicrobial used for the treatment of SD, was diluted ranging from 16 to 1 μg/mL and served as the control. Then, 100 μL of two-fold dilutions of the selected substances was added to 100 μL of the inoculum for each microbial strain to obtain a final concentration of 1.0 × 10^6^ CFU/mL and incubated at 37 °C for 24 h. Microbial growth inhibition was evaluated by comparing the growth of selected microorganisms in wells filled with inoculum without compounds as a positive control and in wells filled with culture medium as a negative control by adding 0.01% resazurin salt solution (Sigma-Aldrich^®®^, St. Louis, MO, USA) [78]. The MICs of the tested substances were determined as the lowest concentration that was capable of inhibiting microbial growth after incubation. The experiments were carried out in triplicate.

### 3.4. Antifungal Activity

#### 3.4.1. Test Microorganisms and Growth Conditions

The fungal strain *M. furfur* 14251 was selected from ATCC. The Sabouraud dextrose agar (SDA) culture medium was used for the initial cultivation of tested fungus at 37 °C for 24 h. Then, *M. furfur* was cultivated in a selective liquid culture medium such as a Sabouraud dextrose broth (SDB) during the experiments.

The 96-well microtiter plates (96 Well EDGE Cell Culture Plates, Nest Scientific Biotechnology, Wuxi, China) were used for consecutive dilutions.

#### 3.4.2. Antifungal Screening by Determining the MICs of Substances

The broth microdilution method was based on the EUCAST rules for antimicrobial susceptibility testing [79] and used for determination of MICs for tested substances. The water solubility of the substances was improved by adding of 0.3% Polysorbate 20 to the initial solutions. Fungal growth controls and sterility controls of the culture medium were performed.

The final concentrations of the substances were 1.00, 0.50, 0.25, and 0.125 weight %. Antifungal drugs such as climbazole ad ketoconazole, recommended for the treatment of SD, were used as controls. The fungal inoculum of 1 mL at a 1.0 × 10^6^ CFU/mL concentration was mixed with 9 mL of each sample concentration prepared in Mueller Hinton Broth. The experimental procedure consists of the substance incubation with a known suspension of *M. furfur* at 37 °C for 24 h [79]. Fungal growth inhibition was evaluated by comparing the growth of selected fungi in wells filled with inoculum without compounds as a positive control and in wells filled with culture medium as a negative control with selective plate count (CHROMagar Malassezia). The MICs of the tested substances were determined as the lowest concentration that was capable of inhibiting growth after incubation. The experiments were carried out in triplicate.

#### 3.4.3. Antifungal Screening by Determining the Log10CFU Reduction

The Log10CFU reduction in fungi in the presence of antifungal substances was used to evaluate the inhibitory effect of the tested substance on a population of *M. furfur* by exposure to the substance through quantitate evaluation of viable total colony counts. The final concentrations of substances were 1.00, 0.50, 0.25, and 0.125 weight % prepared in water with 0.3% Polysorbate 20. Antifungal drugs such as climbazole ad ketoconazole, recommended for the treatment of SD, were used as controls. Then, 1 mL of microorganisms at a 1.0 × 10^6^ CFU/mL concentration was cultured with 9 mL of each sample concentration, prepared in Mueller Hinton Broth. The experimental procedure consisted of the substance incubation with a known suspension of *M. furfur* at 37 °C for 1 h as the average time for a possible effect of drugs for the treatment of SD [79]. The incubation was performed on substrate culture media (Mueller Hinton broth). After the incubation period, the viable fungi population was monitored by counting the viable cells on an agar plate (CHROMagar Malassezia). The suspension of the microorganism inoculum without substance was used as a control. The experiments were carried out in triplicate. The result was expressed in a decimal logarithm of the reduction in colony forming units per mL (log10CFU/mL). The log reduction of more than 2 values corresponded to an antifungal efficacy of more than 99.0%. The experiments were carried out in triplicate.

## 4. Conclusions

The phytochemical composition of the plant-based substance consisting of TTO, 1,8-cineole, and (-)-α-bisabolol was studied using the developed GC/MS chromatography method. The ten antimicrobial monoterpenes and sesquiterpenes specific to TTO and seven new terpenes previously undetected in TTO were identified. Presumably, the last seven phytochemicals were products of a chemical reaction between the main constituents. The plant-based substance demonstrated a high antibacterial and antifungal activity against standard strains of *S. aureus*, *S. epidermidis*, *C. albicans*, and *M. furfur* that are proven to be involved in the pathogenesis of SD. The MIC values of the plant-based substance revealed the synergism of the constituents and allowed the establishment of a targeted antibacterial effect, suggesting a beneficial effect for normal scalp microflora. The antimicrobial activity of the plant-based substance was comparable to benzalkonium chloride, ketoconazole, and climbazole; therefore, this substance is promising for further research and evaluation of possible drug development for the treatment of SD.

## Figures and Tables

**Figure 1 pharmaceuticals-16-00328-f001:**
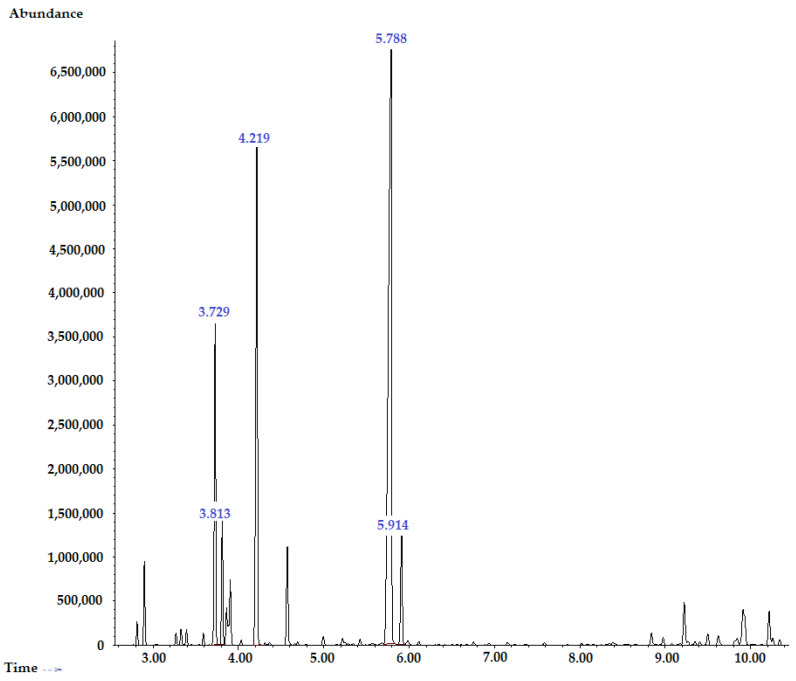
GS/MS chromatogram of TTO. Most of TTO phytoconstituents were thujen, α-pinene, β-pinene, myrcene, α-terpinene, o-cymene, D-limonene, 1,8-cineole, γ-terpinene, terpinolene, terpinen-4-ol, α-terpineol, aromadendrene, viridiflorene (ledene), and δ-cadiene. The identified compounds belonged to the monoterpene and sesquiterpene chemical classes (Table 1).

**Figure 2 pharmaceuticals-16-00328-f002:**
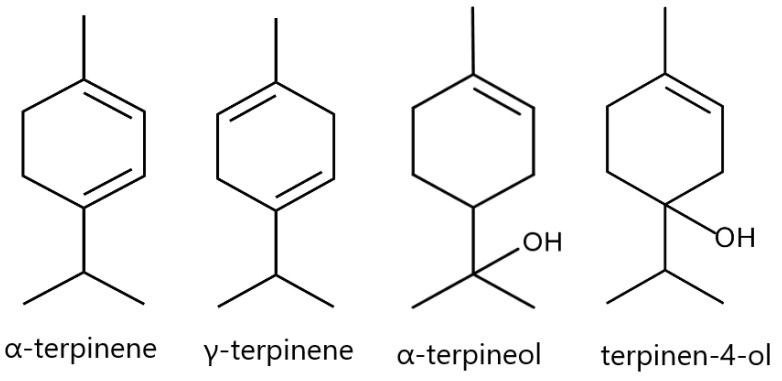
Chemical structures of the major phytoconstituents determined in TTO via GC/MS analysis.

**Figure 3 pharmaceuticals-16-00328-f003:**
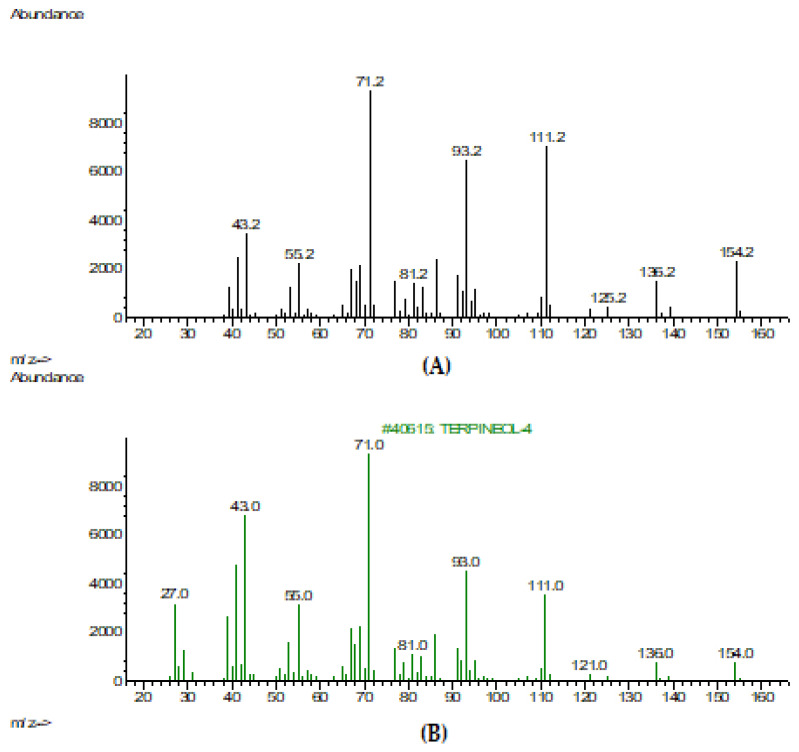
(**А**) Mass spectrum of terpinen-4-ol in TTO; (**B**) Mass spectrum of terpinen-4-ol from the NIST-2017 library.

**Figure 4 pharmaceuticals-16-00328-f004:**
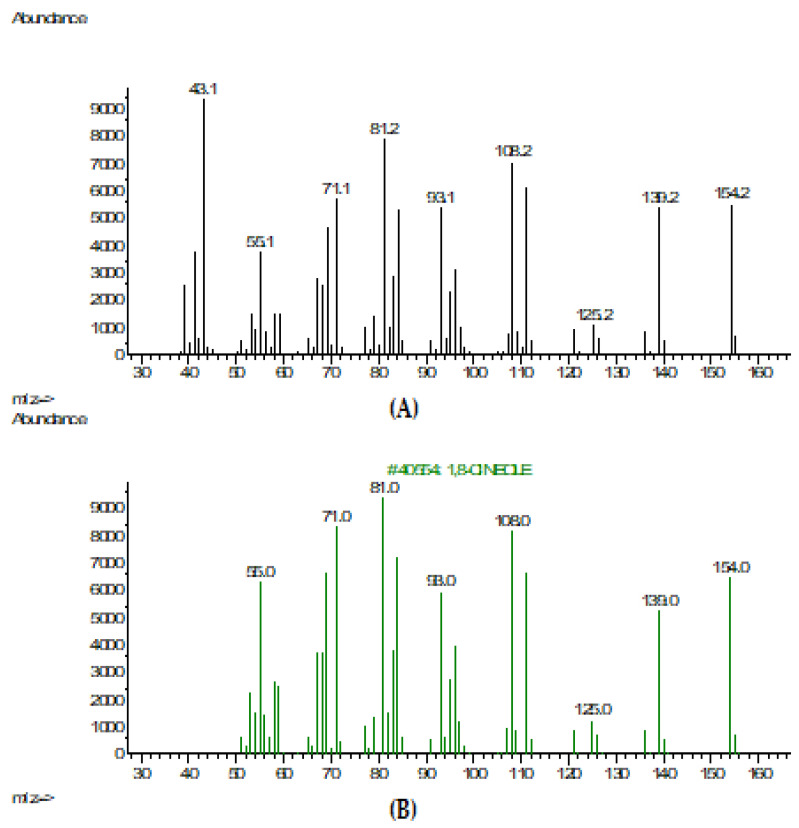
(**А**) Mass spectrum of 1,8-cineole of pure sample; (**B**) Mass spectrum of 1,8-cineole from the NIST-2017 library.

**Figure 5 pharmaceuticals-16-00328-f005:**
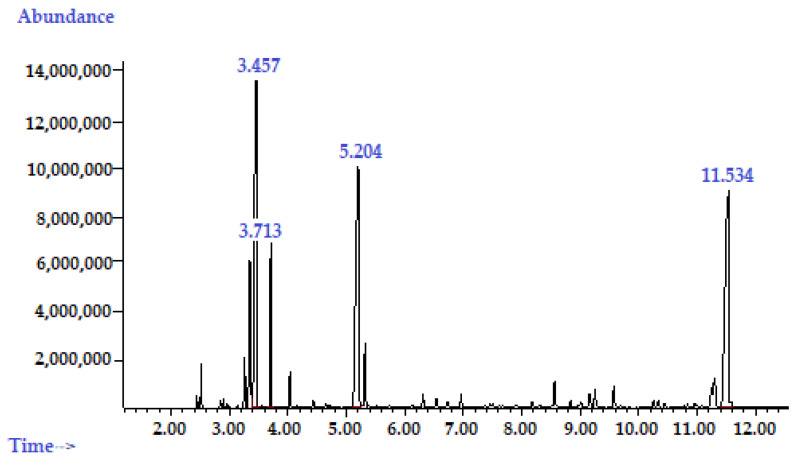
GS/MS chromatogram of the plant-based substance (TTO:1,8-cineole:α-(-)-bisabolol in a 1:1:1 ratio).

**Figure 6 pharmaceuticals-16-00328-f006:**
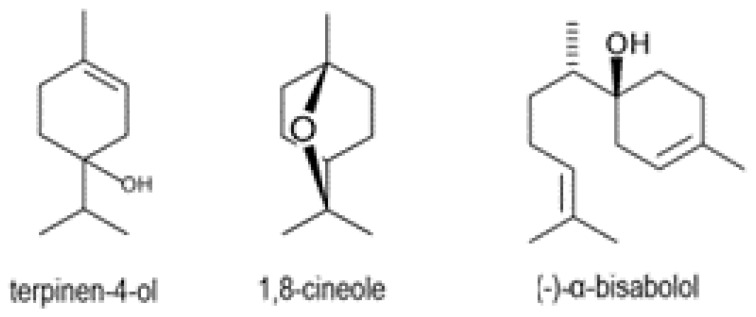
Chemical structures of the major phytoconstituents identified in the plant-based substance via GC/MS analysis.

**Figure 7 pharmaceuticals-16-00328-f007:**
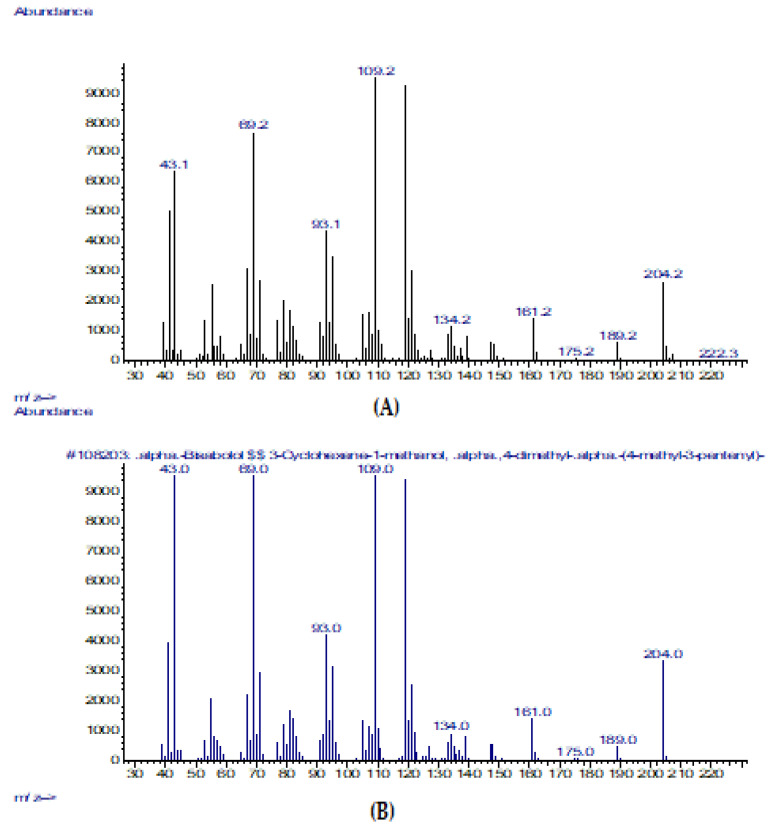
(**А**) Mass spectrum of α-(-)-bisabolol from the plant-based substance (TTO:1,8-cineole:α-(-)-bisabolol in a 1:1:1 ratio); (**B**) Mass spectrum of α-(-)-bisabolol from the NIST-2017 library.

**Table 1 pharmaceuticals-16-00328-t001:** TTO constituents.

No.	Compound ^a^	MW ^b^	Class of Terpenes	Retention Time	RI ^c^
1	α-Thujene	136.23	Bicyclic monoterpene	2.816	924
2	α-Pinene	136.23	Bicyclic monoterpene	2.901	932
3	β-Pinene	136.23	Bicyclic monoterpene	3.331	974
4	Myrcene	136.23	Acyclic monoterpene	3.394	988
5	α-Terpinene	136.23	Cyclic monoterpene	3.729	1014
6	o-Cymene	134.22	Aromatic monoterpene	3.813	1022
7	D-Limonene	136.23	Cyclic monoterpene	3.864	1024
8	1,8-Cineole (eucalyptol)	154.25	Bicyclic epoxy monoterpene	3.907	1026
9	γ-Terpinene	136.23	Cyclic monoterpene	4.219	1054
10	α-Terpinolene	136.23	Cyclic monoterpene	4.575	1086
11	Terpinen-4-ol	154.25	Cyclic monoterpene	5.788	1174
12	α-Terpineol	154.25	Cyclic monoterpene	5.914	1186
13	Aromadendrene	204.35	Sesquiterpene	9.227	1439
14	Viridiflorene (ledene)	204.35	Sesquiterpene	9.914	1496
15	δ-Cadinene	204.35	Sesquiterpene	10.219	1522

^a^ Compounds listed based on their elution order on an HP-5MS column. ^b^ The average molecular weight (MW) of molecules from PubChem and National Institute of Standards and Technology (NIST) Standard Reference Database 69. ^c^ Literature retention index (RI) [42].

**Table 2 pharmaceuticals-16-00328-t002:** The plant-based substance constituents.

No.	Compound ^a^	MW ^b^	Class of Terpenes	Retention Time (Min) ^c^	RI ^d^	Relative Content, %
1	α-Pinene	136.23	Bicyclic monoterpene	2.521	932	1.30%
2	Sabinene	136.23	Bicyclic monoterpene	2.850	969	0.20%
3	β-Pinene	136.23	Bicyclic monoterpene	2.901	974	0.26%
4	α-Phellandrene	136.23	Cyclic monoterpene	2.448	1002	0.32%
5	α-Terpinene	136.23	Cyclic monoterpene	3.264	1014	1.91%
6	o-Cymene	134.22	Aromatic monoterpene	3.355	1022	8.16%
7	1,8-Cineole (eucalyptol)	154.25	Bicyclic epoxydenated monoterpene	3.457	1026	22.28%
8	γ-Terpinene	136.23	Cyclic monoterpene	3.713	1054	6.09%
9	α-Terpinolene	136.23	Cyclic monoterpene	4.041	1086	1.28%
10	1-Methyl-4-(1-methylethenyl)-2-cyclohexene-1-ol	154.25	Cyclic oxygenated monoterpene	4.438	1127	0.36%
11	Terpinen-4-ol	154.25	Cyclic oxygenated monoterpene	5.204	1174	20.89%
12	α-Terpineol	154.25	Cyclic oxygenated monoterpene	5.324	1186	2.96%
13	Trans-ascaridole glycol	170.25	Cyclic oxygenated monoterpene	6.314	1273	1.08%
14	Nonidentified compound	-	-	6.966	-	0.56%
15	Alloaromadendrene	204.35	Sesquiterpene	8.564	1445	1.49%
16	δ-Cadinene	204.35	Sesquiterpene	9.575	1522	1.09%
17	Nonidentified compound	-	-	10.258	-	0.34%
18	Nonidentified compound	-	-	10.340	-	0.46%
19	Trans-caryophyllene	204.35	Bicyclic sesquiterpene	11.245	1531	1.08%
20	(-)-α-Bisabolol	222.37	Monocyclic sesquiterpene alcohol	11.534	1685	25.73%
21	Farnesol	222.37	Acyclic sesquiterpene alcohol	11.301	1698	2.16%
	Total identified (%)					98.64%
	Chemical classes (%)					
	Monoterpene hydrocarbons					19.52%
	Oxygenated monoterpenes					47.57%
	Sesquiterpene hydrocarbons					31.55%
	Nonidentified compounds					1.36%

^a^ Compounds listed based on their elution order on an HP-5MS column (20 m × 0.18 mm i.d., 0.18 μm f.t.). ^b^ The average MW of molecules from PubChem and NIST Standard Reference Database 69. ^c^ Retention time was determined after GC/MS analysis. ^d^ Literature RI [42].

**Table 3 pharmaceuticals-16-00328-t003:** The identification of compounds in the plant-based substance via MS.

No.	Compound ^a^	MW ^b^	Chemical Formula	*m/z* Peaks ^c^	
				Top Peak	Highest Peaks
1	α-Pinene	136.23	C_10_H_16_	93.1	39.1; 53.1; 67.1; 77.1; 105.1; 121.1; 136.2
2	Sabinene	136.23	C_10_H_16_	93.1	41.1; 53.1; 65.1; 77.1; 121.1; 136.2
3	β-Pinene	136.23	C_10_H_16_	93.1	41.1; 69.2; 121.2
4	α-Phellandrene	136.23	C_10_H_16_	93.1	39.1; 65.1; 77.1; 115.0; 136.2
5	α-Terpinene	136.23	C_10_H_16_	93.1	77.1; 91.0; 121.2; 136.0
6	o-Cymene	134.22	C_10_H_14_	119.2	39.1; 65.1; 91.1; 134.2
7	1,8-Cineole (eucalyptol)	154.25	C_10_H_18_O	43.1	58.1; 71.0; 81.2; 93.0; 108.2; 139.2; 154.2
8	γ-Terpinene	136.23	C_10_H_16_	93.1	39.1; 51.1; 65.1; 77.1; 105.1; 121.2; 136.2
9	α-Terpinolene	136.23	C_10_H_16_	93.1	42.1; 79.0; 91.0; 121.1; 136.2
10	1-Methyl-4-(1-methylethenyl)-2-cyclohexene-1-ol	154.25	C_10_H_18_O	43.0	71.0; 93.1; 139.2
11	Terpinen-4-ol	154.25	C_10_H_18_O	71.1	43.1; 93.1; 111.1; 136.2; 154.2
12	α-Terpineol	154.25	C_10_H_18_O	59.1	43.1; 81.1; 93.1; 107.1; 121.1; 136.2
13	Trans-ascaridole glycol	170.25	C_10_H_16_O_2_	109.0	43.0; 9.0; 71.0; 81.0; 127.0
14	Nonidentified compound	-	-	-	-
15	Alloaromadendrene	204.35	C_15_H_24_	41.1; 91; 1; 161.2	67.0; 77.1; 105.1; 119.1; 133.1; 147.2; 175.2; 189.2; 204.2
16	δ-Cadinene	204.35	C_15_H_24_	161.2	41.1; 91.1; 105.1; 119.1; 134.2; 189.2; 204.2
17	Nonidentified compound	-	-	-	-
18	Nonidentified compound	-	-	-	-
19	Trans-caryophyllene	204.35	C_15_H_24_	133.1	41.1; 79.1; 91.1; 93.1
20	(-)-α-Bisabolol	222.37	C_15_H_26_O	109.2	43.1; 69.2; 93.1; 119.1; 134.2; 161.2; 189.2; 204.2
21	Farnesol	222.37	C_15_H_26_O	69.2	41.1; 81.0; 93.1; 121.2; 161.2

^a^ Compounds listed based on their elution order on an HP-5MS column (20 m × 0.18 mm i.d., 0.18 μm f.t.). ^b^ The average MW of molecules from PubChem and NIST Standard Reference Database 69. ^c^ The *m/z* peaks were determined after MS analysis.

**Table 4 pharmaceuticals-16-00328-t004:** MICs of the plant-based compounds and substances against reference microbial strains.

No.	Strain ^b^	MIC (mg/mL) ^a^				
		TTO	1,8-cineole	(-)-α-bisabolol	Plant-Based Substance in a 1:1:1 Ratio	Benzalkonium Chloride
1	*S.* *aureus* ATCC 29213	40.00	40.00	5.00	1.25	5.00
2	*S. epidermidis* ATCC 14990	1.25	2.50	1.25	1.25	5.00
3	*M.* *luteus* ATCC 10240a	40.00	40.00	20.00	20.00	5.00
4	*C. albicans* ATCC 10231	5.00	40.00	20.00	1.25	10.00

^a^ The MIC was determined after 24 h of treatment via broth microdilution method using resazurin. All experiments were carried out in triplicate. ^b^ Bacteria were purchased from American Type Culture Collection (ATCC).

**Table 5 pharmaceuticals-16-00328-t005:** MICs of substances against several fungal strains involved in the pathogenesis of SD.

No.	Strain ^b^	MIC (%) ^a^		
		Plant-Based Substance at a Ratio of 1:1:1	Climbazole	Ketoconazole
1	*M. furfur* ATCC 14251	0.50%	1.00%	0.25%
2	*C. albicans* ATCC 10231	0.25%	1.00%	0.25%

^a^ The MIC was determined after 24 h of treatment via broth microdilution method using resazurin. All experiments were carried out in triplicate. ^b^ Fungi were purchased from ATCC.

**Table 6 pharmaceuticals-16-00328-t006:** Antifungal activity of substances against *M. furfur*.

No.	Strain ^b^	Log10CFU, Mean ± SD ^a^				
		Negative Control	0.125%	0.25%	0.50%	1.00%
1	Plant-based substance at a ratio of 1:1:1	5.85	5.67 ± 0.18	4.42 ± 0.11	3.25 ± 0.13 *	3.04 ± 0.10 *
2	Climbazole		5.08 ± 0.21	5.04 ± 0.14	4.95 ± 0.08	4.92 ± 0.11
3	Ketoconazole		4.22 ± 0.04	<1.00 *	<1.00 *	<1.00 *

^a^ The Log10CFU reduction was determined after 24 h of treatment by counting the viable cells on an agar plate. All experiments were carried out in triplicate. * *p* < 0.05 (ANOVA). ^b^
*Malassezia furfur* was purchased from ATCC.

**Table 7 pharmaceuticals-16-00328-t007:** Components of the tested plant-based substance.

No.	Chemical	Origin	CAS Number	Manufacturer
1	Essential oil of *M. alternifolia*	Leaves of *M. alternifolia*	68647-73-4	Bernardi Group, Grasse, France
2	1,8-cineole (eucalyptol)	Leaves of *Eucalyptus spp*.	470-82-6	Wuxi Lotus Essence Co., Ltd., Wuxi, Jiangsu, China
3	(-)-α-Bisabolol	Leaves of *Hymenocrater yazdianus*	23089-26-1	Merck KGaA, Darmstadt, Germany

## Data Availability

The data presented in this study are available on request from the corresponding author.

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
