# Peer review of "Chemical Composition and Antimicrobial Potential of a Plant-Based Substance for the Treatment of Seborrheic Dermatitis"

_pharmaceuticals, 2023, doi:10.3390/ph16030328_

Round 1
Reviewer 1 Report
First, I would like to learn why the tested organisms were selected just from ATCC collection and no wild or resistant strains were used. This could represent a bias and I would like to understand the view of the authors before my decision.
Please use italics wherever organisms names are written. Please check again all over the document for small typos.
Author Response
Dear Reviewer,
We have prepared answers to your comments. Please see the attachment.

Reviewer 2 Report
The work is original. However, enriching the discussion a little more will reveal the importance of the study better.
Some corrections to the author are below:
Page 1, lines 35-37: the sentence must be corrected, the verb is missing
Page 2, line 79: ...the decrease in TEWL.... What is abbreviation of TEWL? Malassezia spp. (species names must be italicised).
Page 3, line 124: 2.1. GC/MS Analysis of Phytoconstituents from TTO and QC
What is abbreviation of QC?
How did you determine the concentrations of the compounds used for MICs determination?
Why did you only use gram positive bacteria?
Author Response

(The authors gave the same response as above.)

Reviewer 3 Report
The authors state that "the objective of this work was to determine the chemical composition of" Melaleuca alternifolia leaf oil "and to evaluate antimicrobial activity against microorganisms involved in the pathogenesis of SD" (seborrheic dermatitis). The effort involves the use of gas chromatography and mass spectrometry to characterise the components of the complex mixture. This work has been accomplished and the resulting data are extensively tabulated. Minimum inhibitory concentrations are reported for four bacterial species and comparative efficacies are reported for two additional bacteria.
A great deal of detail is presented in the manuscript. However, there was no chemical structure included. From the perspective of an organic chemist, it would be useful to have at least the most abundant compounds illustrated.
My major concern is that the conclusions do not convey to me the essence of the findings. The MIC data show that in each case a well known substance -- not always the same one -- against all of the bacteria. Further, the authors state that the "result has important implications for the human organism, scalp microorganisms such as are part of this microbiota." The implications are less clear to me because there are no toxicity data included. The authors do specify that "more research is needed to elucidate the mechanisms of the action and stability of plant-based substance in the final medical formulations against SD." It seems to me that the mechanism(s) of action is subordinate to a need for dermatological data. I think that a revision of the manuscript needs to include at least a disclaimed if the toxicity or allergenic responses are not known. I also note that a "one hour contact" seems excessive for a potential dandruff treatment
Overall, the experimental work seems to be properly done and the presentations of data, except as noted below, are tedious but satisfactory. I do wonder why retention times and refractive indices are needed in tables, but it is the authors' choice. I do fell that the paper could be improved by some conclusory remarks or generalizations at the end of each section. I think that the actual conclusion could be made more specific.
Minor issues
All organisms should be given their complete names before abbreviations are used.
line 51 i 3% should be in 3%
line 71 perspective should be prospective
line 176 the overlapping numbers on the graph make is difficult to read.
Author Response

(The authors gave the same response as above.)

Reviewer 4 Report
The paper explores the chemical composition and antimicrobial potential of plant-based substances for the treatment of seborrheic dermatitis, but is insufficiently innovative and lacks novelty. All image qualities should be improved.
Author Response

(The authors gave the same response as above.)

Round 2
Reviewer 3 Report
I think that the paper is of average interest and average impact, but it is suitable for publication.
Author Response
Dear Sir,
Thank you for the positive feedback on our article devoted to the determination of the chemical composition and evaluation of the antimicrobial activity of the innovative plant-based substance against standard microorganisms involved in the pathogenesis of seborrheic dermatitis.
Reviewer 4 Report
The author completely answered my question and made great changes to the manuscript according to the comments. In my opinion it should be published in Molecules after addressing some minor comments:
Abbreviations should be defined the first time they appear in each of three sections: the abstract; the main text; the first figure or table
The writing format of species names needs to be unified, such as Malassezia furfur and M. furfur often appears in the same paragraph, lines 423-436.
M. alternifolia should be italic, Table 7
Whether to add spaces before ℃ and % should be unified in the artical
Author Response
Dear Sir,
Thank you for the positive feedback on our article devoted to the determination of the chemical composition and evaluation of the antimicrobial activity of the innovative plant-based substance against standard microorganisms involved in the pathogenesis of seborrheic dermatitis.
The Pharmaceuticals journal was chosen for this publication because the developed plant-based substance based on Melaleuca alternifolia leaf oil, 1,8-cineole and α-(-)-bisabolol has the promising potential for implementation in medical dosage forms for the treatment of seborrheic dermatitis. The search for optimal ratios and synergy for a combination of substances from plant resources is a modern direction in the field of pharmaceutical sciences. We think that our work is in line with the section «Natural Products» and the special issue «Application of Gas Chromatography to Detect Volatile Secondary Metabolites from Natural Products of Pharmacological Interest».
We have checked the text, corrected the abbreviations, and unified the names of microorganisms into Italics in manuscript according to our comments.